# Clinical Application of Bioresorbable, Synthetic, Electrospun Matrix in Wound Healing

**DOI:** 10.3390/bioengineering10010009

**Published:** 2022-12-21

**Authors:** Matthew MacEwan, Lily Jeng, Tamás Kovács, Emily Sallade

**Affiliations:** Acera Surgical, 1650 Des Peres Rd. Ste 120., St. Louis, MO 63131, USA

**Keywords:** extracellular matrix, electrospinning, Restrata, wound healing, soft tissue repair

## Abstract

Electrospun polymeric matrices have long been investigated as constructs for use in regenerative medicine, yet relatively few have been commercialized for human clinical use. In 2017, a novel electrospun matrix, composed of two synthetic biocompatible polymers, polyglactin 910 (PLGA 10:90) and polydioxanone (PDO) of varying pore and fiber sizes (i.e., hybrid-scale) was developed and cleared by the FDA for human clinical use. The present review aims to explain the mechanism of action and review the preclinical and clinical results to summarize the efficacy of the matrix across multiple use cases within the wound care setting, including an assessment of over 150 wounds of varying etiologies treated with the synthetic matrix. Clinical data demonstrated effective use of the synthetic hybrid-scale fiber matrix across a variety of wound etiologies, including diabetic foot and venous leg ulcers, pressure ulcers, burns, and surgical wounds. This review represents a comprehensive clinical demonstration of a synthetic, electrospun, hybrid-scale matrix and illustrates its value and versatility across multiple wound etiologies.

## 1. Introduction

Acute and chronic wounds can present significant challenges for patients and wound care professionals, impacting quality of life and total cost of care [1]. The process of wound healing, especially in refractory wounds, can be challenging, and requires a coordinated effort of cellular recruitment, neovascularization, and tissue ingrowth, which in many wounds is disrupted by tissue loss and/or underlying pathology [1]. In these instances, regenerative matrix materials have been used to provide immediate wound coverage and support cellular infiltration and wound healing, with the hope of generating viable tissue [1]. Various biologic and synthetic solutions have been utilized clinically, including human autografts and allografts, animal-based xenografts, skin substitute materials, and fully synthetic matrices [1]. A sample of available wound healing matrices is shown in Table 1. Autograft availability is limited and creates iatrogenic morbidity at donor sites. Allografts and xenografts eliminate the morbidity associated with autografts but introduce the need for decellularization, additional risks of inflammatory response and disease transmission, and challenges associated with storage and handling, such as cold storage and tissue tracking. Alternative approaches, including negative pressure wound therapy [2], platelet derived growth factor therapy [3], and platelet rich plasma therapies [4] have also been investigated as solutions for refractory wounds.

Electrospun polymeric matrices have long been investigated as an innovative construct for use in tissue engineering and regenerative medicine research due to their ability to mimic the structure and scale of native tissue [5]. Solvent-based electrospinning offers a unique method of forming non-woven, fibrillar materials composed of a variety of biocompatible and bioresorbable polymer fibers [6]. During electrospinning, a high voltage electric field is being utilized to create a fiber jet between a needle connected to a charged polymer solution source and a collector where newly formed fibers will be deposited. Adjusting many parameters in the electrospinning process provide the capability to adjust the size, distribution, and morphology of fibers [7]. Recent advancements have shown that the electrospinning technique can be combined with other scaffold fabrication techniques, including 3D printing, to develop matrices with tailored properties for tissue regeneration [8,9].

Biomimetic features are critical to an electrospun material’s mechanism of action, and specifically designing a matrix to mimic the scale and structure of native tissue has been proven to yield benefits in the repair of soft tissue defects [10]. For instance, a biomaterial’s composition and structure can have a significant effect on its angiogenic capacity, as the architecture of a scaffold will impact its ability to allow for transport of blood supply as well as cells and nutrients [10]. Fibroblasts have been shown to proliferate and spread more rapidly and upregulate collagen expression on poly(lactic-co-glycolic acid) (PLGA) fibers 350–1100 nm in diameter, as compared to fibers of smaller or larger dimensions [11]. This process of contact guidance, supported via topographical cues provided by the electrospun fibers, is mechanistically distinct from that of other xenogenic and human allogenic matrices which operate largely via transient delivery of trophic factors [12]. In addition to sub-micron fibers that encourage the rapid spread of fibroblasts, a matrix that also contains larger micron-scale fibers will benefit from increased space between sub-micron scale fibers, opening the material’s morphology, and thus creating an electrospun matrix with an interconnected pore structure that is similar to natural tissue and supportive of cellular migration and tissue ingrowth as well as neovascularization [13].

Porosity is also an important feature of any electrospun scaffold, and different pore dimensions will encourage different regenerative responses [14]. It has been shown, for instance, that the optimal pore size for neovascularization is 5 μm, while for fibroblast ingrowth, a pore size of 5–15 μm is optimal. Skin regeneration calls for a larger range of pore sizes at 20–125 μm [14]. A pore size of 1–10 μm has been shown to prevent bacterial penetration [15]. Therefore, a biodegradable electrospun matrix possessing a range of fiber sizes (from a few hundred nanometers to a few microns), and a distribution of pore sizes (from 5–125 μm), may be well suited for neovascularization, fibroblast ingrowth and retention, and prevention of bacterial penetration in the early stages of use. Over time, as the fibers gradually degrade the porosity will open further, allowing for additional tissue ingrowth and skin regeneration. The electrospun, synthetic, hybrid-scale fiber matrix is engineered to resorb at a rate that matches that of cellular ingrowth, and typically degrades via hydrolysis in about 1–3 weeks dependent on the wound bed [16]. This controlled rate allows for mechanical offloading from the matrix to the newly formed tissue [16]. Degradation rate is vital to the porosity of the matrix, and therefore directly correlates to the infiltration and proliferation of cells within the matrix. As such, fibers such as PGLA (10:90) and PDO were utilized in the matrix as opposed to fibers such as PLLA and PCL, which degrade via hydrolysis at a significantly slower rate [16]. Thus, electrospun materials, specifically those that possess a range of fiber diameters and pore sizes that mimic the hierarchical organization of native tissue (i.e., hybrid-scale fiber matrices), are able to strike the right balance of a rapid response and longevity in order to heal intractable wounds unseen in polymer scaffolds composed utilizing alternative processing methods [13,17].

Aside from possessing an architecture similar to native tissue, electrospun hybrid-scale fiber matrices can be engineered to incorporate additional properties that further add value in the wound care setting (Figure 1).

Fully synthetic materials are resistant to rapid enzymatic degradation typical of collagen-based materials, which degrade prematurely due to proteases that are overexpressed in chronic non-healing wounds, such as matrix metalloproteinases (MMPs) [17]. As collagen materials degrade, the also negatively impact cellular infiltration and differentiation [17]. Both alginate and collagen materials have been observed to stimulate immune and inflammatory responses, which subsequently can negatively impact wound healing [17,18]. As such, synthetic matrices avoid premature degradation, and thereby the need for frequent reapplication is reduced [17]. Furthermore, monomers and degradation products of bioresorbable polymers utilized to fabricate electrospun matrices have been shown to have antimicrobial effects; or instance, lactic acid and polylactic acid have been shown to have an antimicrobial effect across multiple bacterial strains [19,20,21,22,23,24,25]. Moreover, wounds that heal have been found to have lower pH (acidic), similar to intact skin, which is naturally acidic, in contrast to nonhealing wounds, which tend to have higher pH (alkaline) [22,24,25]. Therefore, the acidic environment induced by acidic components may help support wound healing by resisting bacterial growth and biofilm formation and restoring tissue pH to normal mildly acidic levels [19,20,21,22,23,24,25]. Use of synthetic polymers also offers the ability to engineer a wound matrix material that may minimize the risk of allergic response to animal products, zoonotic disease transmission, or ethical or religious concerns.

Despite these many advantages, electrospun hybrid-scale fiber matrices have not been commercialized on a wide scale for human clinical use. Electrospun nanomaterials have been investigated in the wound healing space in conjunction with gelatin, honey, and antioxidants, however none have been explored in vivo, and some have been observed to degrade at a rapid rate that does not facilitate wound healing [26,27,28]. The investment in both time and cost to develop a product that meets the Food and Drug Administration’s (FDA) standards for human use often limits the transition of these materials from a research setting to a commercial one. To commercialize materials for human use in the United States, the FDA requires that quality standards be met throughout the design, development, and manufacturing phases of product development. Additionally, there is the need for safety and efficacy data, scaled manufacturing techniques that meet rigorous process validation standards, and the establishment of a reliable distribution channel. Despite these challenges, our group translated these aforementioned design features into a commercially available electrospun hybrid-scale fiber matrix indicated for treating acute and chronic wounds (tradename: Restrata^®^). The synthetic hybrid-scale fiber matrix possesses a fibrous structure of varying fiber diameters with high porosity [17]. The architecture of the material, which is similar to native extracellular matrix, allows for cell ingress and retention, as well as neovascularization of the newly forming tissue, before completely degrading via hydrolysis [17]. The electrospun matrix is composed of two synthetic biocompatible polymers, polyglactin 910 (PLGA 10:90) and polydioxanone (PDO), both of which are used in other (FDA)-cleared devices, including dura substitutes, resorbable sutures, and orthopedic implants [17]. PLGA 10:90 and PDO were specifically selected in order to achieve a matrix with optimal handling properties and a rate of resorption ideally matched to the process of new tissue formation and wound healing. These polymers are co-electrospun into soft, white, compliant (i.e., a strip of the material will drape over a user’s finger under its own weight at room temperature) non-woven sheets capable of supporting the natural wound healing process [17]. Once applied to a wound, the matrix supports cellular infiltration, new tissue formation, and wound healing while progressively resorbing into the tissue over the course of 2 weeks, on average.

Regulatory clearance of the synthetic hybrid-scale fiber matrix was achieved as a medical device via the premarket notification (510(k)) pathway, for use in the local management of wounds, including partial and full thickness wounds, pressure sores/ulcers, venous ulcers, diabetic ulcers, chronic vascular ulcers, tunneled/undermined wounds, surgical wounds, traumatic wounds, partial thickness burns, and draining wounds. The clearance of the hybrid-scale fiber matrix has led to multiple first-in-man demonstrations of the efficacy and value of the electrospun material in the wound care treatment setting.

The present review aims to highlight the preclinical and clinical evaluation of one synthetic electrospun hybrid-scale fiber matrix and summarize the clinical efficacy of the matrix across multiple use cases within inpatient and outpatient treatment settings. Given that this is the first electrospun hybrid-scale fiber matrix commercialized for in human use, the authors feel it is necessary to explore the various clinical uses and observed outcomes.

## 2. Material Fabrication and Methods

The electrospun hybrid-scale fiber matrix is fabricated by solvent electrospinning polyglactin 910 (melt temperature 195–230 °C) and polydioxanone (inherent viscosity 1.5–2.2 dL/g) into compliant three-dimensional (3-D) wound matrix (Restrata^®^, Acera Surgical, Inc., St. Louis, MO, USA). The resorbable polymers create an electrospun matrix consisting of fibers of various sizes (Figure 2). Sheets of the electrospun matrix are cut into various sizes, packaged in nested pouches, and sterilized via e-beam radiation.

## 3. Preclinical Results

The safety and efficacy of the electrospun hybrid-scale matrix in the wound care setting was thoroughly evaluated utilizing multiple in vitro and in vivo assays of bioresorbability and biocompatibility and an in vivo model of wound healing.

### 3.1. Biocompatibility Studies

Laboratory testing was conducted per well-accepted International Organization for Standardization (ISO)-10993 standards to assess and confirm the biocompatibility of the synthetic electrospun matrix, including cytotoxicity elution assay, guinea pig sensitization assay, intracutaneous irritation reactivity assay, hemolysis assay, in vitro mouse lymphoma assay, in vivo mouse micronucleus assay, Ames assay for bacterial mutagenicity, rabbit pyrogen assay, acute systemic toxicity assay, sub-chronic and chronic toxicity animal studies, and endotoxin testing. The results demonstrated that the synthetic matrix has excellent biocompatibility properties and is non-toxic, non-pyrogenic, non-genotoxic, non-hemolytic, and non-mutagenic [29].

### 3.2. pH Study

Benchtop testing was performed to measure the effect of dissolution of the synthetic hybrid-scale fiber matrix on local pH over time. Samples of the electrospun matrix were immersed in phosphate buffered saline, and the pH of the solution was measured once a day for 28 days. Results demonstrated that the synthetic matrix reduced the pH in the surrounding media and elicited an acidic microenvironment with pH levels similar to that found in both normal healthy skin and wounds that heal, suggesting that the synthetic matrix may help create an environment that is bacteriostatic and more conducive to wound healing.

### 3.3. Large Animal Study

The synthetic hybrid-scale fiber matrix was evaluated in a clinically relevant porcine full-thickness wound model [30]. Full-thickness wounds 3 cm in diameter were created along the dorsum, between the shoulder and ilium, through the subcutaneous layer to fascia, and the synthetic hybrid-scale fiber matrix (Restrata^®^) or a leading bilayer xenograft matrix (Integra^®^ Bilayer Matrix Wound Dressing, Integra, Plainsboro, NJ, USA) was applied directly to each wound. Wounds were monitored for up to 30 days post-application to assess wound healing and the quality of new tissue formation [30]. Tissue samples were collected, processed for histology, and stained using hematoxylin and eosin (H&E). Histology results found that the synthetic matrix had less inflammation and faster filling of the wound bed with granulation tissue when compared to the xenograft matrix (Figure 3). Overall, the synthetic hybrid-scale fiber matrix demonstrated a faster rate of healing, decreased inflammation, and more complete re-epithelialization and wound closure compared to the bilayer matrix, suggesting that the synthetic matrix could offer a new option for treating wounds and justifying further investigation in the clinical setting [30].

## 4. Clinical Assessments/Studies

Multiple clinical studies, including retrospective analyses, prospective analyses, and case reports have been conducted to evaluate the clinical efficacy and economic benefit of the synthetic hybrid-scale fiber matrix across multiple use cases in the wound care setting. A summary of the clinical studies can be found in Table 2, and details can be found in the subsections below. Assessment of over 150 treated wounds of varying etiologies demonstrated that significant wound healing was observed following treatment with the synthetic hybrid-scale fiber matrix across multiple clinical use settings. Results further demonstrate that the electrospun hybrid-scale fiber matrix also offered unique clinical versatility in facilitating both tissue granulation and a bridge to skin grafting as well as definitive wound closure and re-epithelialization, unlike alternative synthetic scaffolds and matrices.

### 4.1. Chronic Wound Care

A retrospective case series examined 82 chronic wounds treated with the synthetic hybrid-scale fiber matrix. The nonhealing wounds were of varying etiologies, including diabetic foot ulcers (DFUs), venous leg ulcers (VLUs), pressure ulcers (PUs), and postsurgical wounds. The wounds were debrided, the synthetic matrix was applied to the wound bed, and dressings were applied. Wounds were observed weekly, and the synthetic matrix was re-applied as needed. Significant wound healing was observed, with 85% of the treated wounds achieving complete closure at 12 weeks. Of note, VLUs had an impressive closure rate of 91% at 12 weeks [31].

A 3-patient case series of 4 recalcitrant neuropathic foot ulcers reported on the use of the synthetic hybrid-scale fiber matrix followed by subsequent treatment with adjunctive therapy as deemed appropriate. To apply the synthetic matrix, the wounds were debrided, the matrix was fenestrated and applied to the wound bed and covered with dressings. Wounds were observed weekly, and the synthetic matrix was re-applied as needed. For 2 of the patients, adjunctive therapy was used after 6–7 weeks of treatment using the synthetic matrix. The study found that all 4 ulcers achieved significant granulation tissue formation, including coverage over previously exposed bone, and reduction in wound area, with complete wound healing observed for 3 of the ulcers. Use of the synthetic matrix enabled successful limb salvage and avoided the need for amputation [31].

A 23 patient retrospective case series was conducted at a single site. Wound types included TMAs, DFUs, pressure ulcers, partial ray amputations, and VLUs. The synthetic hybrid-scale fiber matrix was prepared by fenestration with either a STSG graft mesher or a #15 blade scalpel. After being cut to the wound size, the matrix was then applied to the wound bed using either sutures or butterfly bandages. Dependent on the wound type, closure occurred either over the synthetic hybrid-scale fiber matrix (in wounds such as TMAs), or the matrix was dressed utilizing non-adherent gauze or bolster dressings. Additionally, some wounds were also treated with NPWT or STSG. Treatment with the synthetic hybrid-scale fiber matrix resulted in complete closure in 96% of wounds, including wounds with exposed joint capsules [33].

A prospective, single-arm clinical study conducted at 5 centers across the United States assessed clinical outcomes of 24 nonhealing DFUs treated with the synthetic hybrid-scale fiber matrix. Wounds were debrided as needed, and the synthetic matrix was fenestrated, cut to size, and secured to the wound bed using sutures, adhesive strips, or staples as desired. Wounds were observed weekly, and the synthetic matrix was re-applied as needed (Figure 4). Overall, the study reported a high incidence (75%) of complete wound closure at 12 weeks. The average time to closure was 6.4 ± 2.5 weeks, and no itching and little to no odor was noted at the wound sites throughout the course of treatment [34].

In a case series, various difficult-to-heal chronic wounds, a VLU, a DFU, a Charcot foot deformity, and two pressure ulcers, were treated using the synthetic hybrid-scale fiber matrix. The results demonstrated excellent healing responses including granulation tissue formation and wound size reduction in all wounds, tissue coverage over the exposed bone in 3 wounds, wound exudate reduction in 2 cases, and infection management in 4 patients. This investigational use of the synthetic hybrid-scale fiber matrix indicated the versatility of this material to be used across a wide range of refractory chronic wounds [35].

To evaluate the efficacy of the synthetic hybrid-scale fiber matrix in augmented flap reconstruction, treatment results of 11 patients with complex pressure ulcers were retrospectively collected. Ulcers were prepared by sharp debridement of non-viable tissue. Next, the synthetic matrix was fenestrated, secured to the wound bed using sutures or staples, and was left in the wound bed to encourage granulation tissue formation prior to flap reconstruction. Patient follow-up conducted up to 165 days post-flap reconstruction indicated an excellent wound closure rate of 90.9% (10 cases of complete wound closure and one case of 97.2% wound area reduction). Moreover, no complication was reported over the course of the treatment. This study suggests that pre-conditioning complex pressure ulcers with the synthetic hybrid-scale fiber matrix improves the efficiency and success of flap coverage and complete ulcer closure [36].

In a retrospective 20-patient study, 18 DFUs and 5 VLUs with the average wound age of 16 months, 78% of which failed one or multiple previous advanced wound therapies and considered difficult-to-heal ulcers with high failure risk for future therapies, were treated using synthetic hybrid-scale fiber matrix (Figure 5). Treatment of these ulcers with the synthetic matrix resulted in 96% healing rate (100% healing of VLUs and 94% healing of DFUs). VLUs healed with an average of 10.8 ± 5.5 applications over 244 ± 153 days, and DFUs healed with an average of 6.7 ± 3.9 applications over 122 ± 69 days. The positive healing response achieved in this study suggested that the synthetic hybrid-scale fiber matrix may provide the critical and needed solution for the chronic difficult-to-heal ulcers [37].

A retrospective case series of 9 patients with multiple severe co-morbidities was conducted at a single site [44]. The patients included in this study underwent amputations of the lower extremity. Amputation types included transmetatarsal, Lisfranc, and partial ray amputations. The synthetic hybrid-scale fiber matrix was fenestrated, cut to wound size, and secured to the wound bed utilizing either sutures or staples. All wounds were also treated with NPWT, and at the clinician’s discretion 5 wounds were also treated with amniotic tissue, 1 was treated with a STSG, and 2 wounds were treated with both amniotic tissue and STSG. All wounds received 1 application of the synthetic hybrid-scale fiber matrix, and 78% went on to achieve complete wound closure. This retrospective series demonstrates the versatility of the matrix in conjunction with other advanced treatment modalities and is especially important amongst a patient population with multiple co-morbidities [38].

In a prospective study, 10 patients who underwent transmetatarsal amputation (TMA) were treated using an augmented closure approach including the application of the synthetic hybrid-scale fiber matrix. In this method, a synthetic matrix was placed in the wound bed to encourage granulation tissue formation, and the wound was closed using sutures. To ensure lasting wound closure and to avoid wound dehiscence, a strip of the synthetic matrix was secured on the suture line (Figure 6). The treatment results of the augmented closure group (experimental group) were compared with 10 patients who underwent the primary closure with suture only (control group). Results showed 80% healing in the experimental group compared to 60% for the control group. Therefore, using the augmented closure approach led to a 20% better limb salvage rate and avoided amputation. Moreover, the augmented closure method resulted in shorter healing duration, lower incidence of wound dehiscence, and less required additional procedures prior to complete healing [39].

### 4.2. Post-Mohs Wounds

A retrospective study reported the use of the synthetic hybrid-scale fiber matrix to reconstruct post-Mohs surgical wounds on the auricular helix of 4 patients (Figure 7). The synthetic matrix was applied to the wound site and was covered by a dressing. Wounds with an average size of 11.8 cm^2^ received 1.25 ± 0.50 applications of the synthetic matrix which led to 100% epithelialization of all wounds in 7.9 ± 4.2 weeks with no reported complications. Evaluation of the healed wounds demonstrated excellent aesthetic results with no scar formation and no skin deformity [40].

### 4.3. Tendon Healing

A 12-patient case series reported on the use of the synthetic hybrid-scale fiber matrix to augment peroneal tendon repairs. Augmenting peroneal tendon repairs with the synthetic hybrid-scale fiber matrix may help minimize complications associated with direct repair, such as reducing painful adhesions, increasing tendon excursion, and facilitating a rapid return to normal function [45]. MRIs were obtained prior to surgery which revealed soft tissue and tendon injury. Patients underwent surgical treatment to repair the tendon using direct repair with suture followed by wrapping of the tendon with the synthetic matrix (Figure 8). The study found that surgical treatment resulted in significant pain relief, with 92% of the patients reporting little to no pain, with associated return to normal function within 2–5 months [41].

### 4.4. Traumatic Wounds

A case series evaluated the use of the synthetic hybrid-scale fiber matrix in place of split-thickness skin graft (STSG) to treat complex skin lesions, including those resulting from calciphylaxis, an abdominal fistula, and necrotizing fasciitis of the hand and arm. The synthetic matrix was fenestrated, used in conjunction with negative pressure wound therapy (NPWT), and re-applied as needed. In all cases, significant healing was observed. Of particular note, the synthetic matrix remained viable even upon exposure to bile output that destroyed other therapies, including a STSG. Preliminary cost analysis also found that the synthetic hybrid-scale fiber matrix resulted in less operating room and surgery time and lower total cost of care compared to alternative treatment options, such as STSGs and amniotic and placental allograft tissues [42].

A separate case report evaluated the use of the synthetic hybrid-scale fiber matrix in conjunction with STSG to treat a traumatic crush injury. The synthetic matrix was first fenestrated, cut to size, and applied to the wound with NPWT. After 3 weeks, the wound bed was deemed to be well-granulated, and a STSG was then applied. After another 3 weeks, successful clinical results and 100% take of the STSG were observed [43].

### 4.5. Other Wounds

A retrospective 2 patient case series of 2 trauma-induced hematomas on the lower extremity investigated the use of the synthetic hybrid-scale fiber matrix in this clinical setting. Both hematomas were evacuated, and the synthetic hybrid-scale fiber matrix was fenestrated and applied to the wound bed. The average number of applications was 2.5 and both wounds achieved complete closure at an average of 11 weeks post-operatively [44].

## 5. Discussion

Despite the longstanding investigation of electrospun matrices as a novel construct for tissue engineering and regenerative medicine applications in an academic setting, little human clinical data is available to confirm the value of the technology in clinical and surgical settings. A limited number of electrospun scaffold products have been successfully translated into human clinical use due to the challenges associated with commercialization, including the time and cost associated with product development under FDA design controls, implementation of commercial scale manufacturing, regulatory clearance activities and product distribution. Despite these challenges, recent work by our group has demonstrated the successful design and clinical translation of a novel electrospun hybrid-scale fiber matrix for use in wound management [17,30,31,32,33,34,35,36,37,38,39,40,41,42,43,44]. Since receiving FDA clearance in 2017, this electrospun matrix has been utilized in multiple clinical and surgical settings and a variety of different wound types and clinical indications [30,31,32,33,34,35,36,37,38,39,40,41,42,43,44]. The present review demonstrates the use of a synthetic hybrid-scale fiber matrix in clinical practice and offers collective insight into the efficacy and value of a hybrid-scale fiber matrix across multiple use cases in the wound care and soft tissue repair setting.

The range of clinical data presented confirms the versatility of the synthetic hybrid-scale fiber matrix across multiple indications. Retrospective and prospective clinical studies demonstrate clinical efficacy of the electrospun matrix in treating chronic wounds, such as chronic diabetic foot ulcers and venous leg ulcers, while retrospective studies and case reports also demonstrate clinical efficacy of the matrix in treating acute wounds, such as post-Mohs wounds and surgical wounds [30,31,32,33,34,35,36,37,38,39,40,41,42,43,44]. These findings suggest that the hybrid-scale fiber matrix provides significant physiologic benefit to wounds across a variety of underlying pathologies, including wounds that may have failed alternative advanced therapies. Positive clinical results in these refractory wounds further indicate that the unique mechanism of action provided by the synthetic hybrid-scale fiber matrix may provide an alternate course of therapy distinct from that of current human allogenic or xenogenic biologic matrices and skin substitutes.

Additionally, the data presented confirmed the multiple roles that the hybrid-scale fiber matrix can play in the wound care continuum. Retrospective clinical data confirm the success of the matrix in achieving definitive wound closure and re-epithelialization when serially applied to stalled or refractory chronic wounds [37]. Furthermore, data supports the role of the hybrid-scale fiber architecture in promoting granulation tissue and preparation of the wound bed, even in the setting of exposed underlying structures, for staged closure via split-thickness skin graft or rotational flap [36,38,39,40]. Clinical evidence also suggests a role for the hybrid-scale fiber matrix in replacing split-thickness skin grafts in achieving definitive wound closure, thereby reducing complications around donor site morbidity and post-operative pain. These findings suggest that hybrid-scale fiber matrices have multiple roles within the wound care continuum and offer versatility in use beyond those of other synthetic matrices. For example, polyurethane temporalizing matrices (NovoSorb BTM, PolyNovo, Port Melbourne, Australia) are limited to use in staging procedures due to the presence of an impermeable silicone backing and required delamination [46], while solvent case polyvinyl alcohol matrices containing silver particles (MicroLyte AG, Imbed Bioscience, Madison, WI, USA) are unable to achieve granulation and volumetric tissue formation needed in deep wounds [47].

Reported clinical results around the treatment of chronic diabetic foot ulcers (DFUs) suggest that the electrospun matrix presents a viable alternative to both standard of care and advanced biologic therapies. A meta-analysis examining healing rates in DFUs treated with standard care methods (offloading and saline or placebo-gel moistened gauze) reported that only 24% of the wounds were healed after 12 weeks [48]. A clinical trial evaluating 154 DFUs treated with a xenogenic bilayer wound matrix (Integra^®^ Dermal Regeneration Template, Integra, Plainsboro Township, NJ, USA), reported closure of only 51% of wounds within 16 weeks [49]. In a randomized clinical trial (RCT), DFUs treated with standard of care or with a bi-layered skin substitute (BLSS, Apligraf^®^, Organogenesis, Canton, MA, USA) in addition to standard of care observed closure of 26% of the control wounds and 52% of the BLSS-treated group at 12 weeks [50]. A prospective study that evaluated BLSS and a cryopreserved split-thickness skin allograft (STSA, TheraSkin^®^, Misonix, Farmingdale, NY, USA) found that 41% of the DFUs treated with BLSS and 67% of the wounds treated with STSA closed within 12 weeks [51]. In another RCT, patients with DFUs were treated human fibroblast-derived dermal substitute (HFDS, Dermagraft^®^, Organogenesis, Canton, MA, USA) in addition to the standard regimen for up to 12 weeks and demonstrated closure of 30% of the HFDS-treated wounds [52]. In a separate study, DFUs treated with a human cellular repair matrix (h-CRM, Grafix^®^, Osiris Therapeutics, Inc, Columbia, MD, USA) demonstrated closure of 85% of the wounds at 12 weeks, with an average time of 6.2 ± 2.6 weeks to complete wound closure [53]. Comparing these healing rates, which range from 24% to 85%, to the clinical results achieved utilizing the synthetic hybrid-scale fiber matrix (in which 34 chronic DFUs were treated and complete closure of 85% of the wounds was observed at 12 weeks) confirm the positive effect of the presence of the electrospun matrix in the wound bed and the associated improvement in successful wound healing imparted by the material [31]. These results suggest that the synthetic hybrid-scale fiber matrix offers, at minimum, a comparable alternative to existing stand of care and long-standing biologic skin substitute products in the setting of chronic DFUs.

Reported clinical results around the treatment of chronic VLUs also suggest that the electrospun matrix meets or exceeds outcomes associated with existing wound care therapies. Reported retrospective data of 34 chronic VLUs demonstrated healing of 91% of the wounds at 12 weeks, highlighting the ability of the electrospun matrix to heal difficult chronic wounds commonly challenged by highly exudative conditions and overexpression of MMPs [31]. An additional retrospective case series evaluating 5 VLUs refractory to advanced wound therapies demonstrated eventual healing of 100% of the wounds [37]. Comparatively, a healing rate of only 60% at 12 weeks was achieved when VLUs of 52 patients were treated using dehydrated amnion/chorion membrane, though wounds exhibited positive percent area reduction as early as 4 weeks (EpiFix, MiMedx, Marietta, GA, USA) [54,55]. Human fibroblast derived dermal substitute (Dermagraft, Organogenesis, Canton, MA, USA) applied in conjunction with compression therapy were also demonstrated achieve healing of only 34% of patients treated by 12 weeks compared with 31% in the control group. These findings highlight the fact that the efficacy of biologic human allogenic product may be limited in the setting of chronic VLUs and that specific biologic options may not provide significant benefit over standard of care [56]. Together these findings support the clinical benefit of the electrospun hybrid-scale fiber matrix and suggest that the unique engineered fully synthetic design may provide additional benefit in the setting of chronic VLUs. Resistance of synthetic matrices to enzymatic degradation, such as that caused by MMPs, and increased longevity in the wound bed may be attributable for this success [17].

Collected clinical data around the hybrid-scale fiber matrix also confirms the clinical efficacy of the material across a number of additional wound types and clinical uses. Evaluation of the electrospun matrix in augmented closure of 11 complex pressure ulcers demonstrated complete closure of 90.9% of wounds after follow-up ranging 31 to 111 days despite multiple comorbidities in studied subjects [36]. Comparably, a 55% healing rate at 12 weeks was reported when PUs of 62 patients were treated with an extracellular matrix graft (OASIS Wound Matrix, Smith and Nephew, Fort Worth, TX, USA) [57]. Documented use of the electrospun matrix as a bridge to definitive closure via split thickness skin graft further highlights the clinical versatility of the engineered material. Demonstration of wound bed granulation within 3 weeks of application of the eletrospun matrix and 100% take of split thickness skin grafts [43] demonstrate comparable performance to bilayer silicone-bovine collagen matrix reporting a mean time to grafting of 26.5 days and mean percentage take of STSGs of 86.1% at 6 months [58]. The hybrid-scale fiber matrix may therefore improve upon documented clinical performance associated with biologic options that may facilitate functional recovery yet result in a lengthy course to complete healing.

Reported clinic data also suggests new use cases for the synthetic hybrid-scale matrix and areas in which the material may provide a new solution to existing clinical needs. A report highlighting the use of the electrospun matrix as a replacement for a harvested split thickness skin graft presents a unique approach to obtaining definitive closure of wounds in patients ineligible for skin grafting procedures [42]. Although limited data from 3 patients has been collected to date, preliminary cost analysis demonstrated that the use of the synthetic hybrid-scale fiber matrix may also result in lower total cost of care compared to alternative treatment options, such as STSGs and amniotic and placental allograft tissues [42]. Additionally, a 20 patient clinical case series presents the use of the electrospun matrix as a means of treating surgical wounds associated with transmetatarsal amputations and potential reduction of post-operative complications including limb loss [39]. Results of the study demonstrated that application of the electrospun matrix resulted in 80% wound healing compared to 60% with primary closure, and a 20% improvement in limb salvage rates post-operatively [39]. Use of the electrospun matrix also resulted in a shorter healing time, lower incidence of wound dehiscence, and fewer procedures prior to complete healing [39]. These results improved outcomes compared with prior clinical studies of TMA wounds that demonstrated a 46% healing rate post-TMA, with a mortality rate of 17% [59]. Similarly, healing rates of 53% and 54% have also been reported following TMA utilizing primary closure without the use of adjunctive therapies [60,61]. These results suggest that many additional applications and uses for the synthetic hybrid-scale matrix may exist beyond those originally anticipated and that additional clinical and economic benefits may be attributable to use of the electrospun matrix in a variety of clinical settings.

Nanofiber applications and electrospun products have been explored to some extent in biomedical applications [26,27,28]. In prior reviews, electrospun nanofibers have demonstrated antioxidant properties in synthetic materials [26]. This is a beneficial property across a variety of clinical applications of biomedical devices, particularly in wounds. Proper medical management of both infected and non-infected wounds has included antioxidant compounds, due to their positive affect on proliferating cells and neovascularization [26]. The synthetic hybrid-scale fiber matrix discussed here as shown a mechanism of action similar to this, by encouraging increased neovascularization and cellular infiltration into the wound bed, in addition to demonstrating an antimicrobial effect as it degrades [17]. The microenvironment of wounds is vital to address in order to impact sustained wound healing [27,28]. Nanofiber and electrospun materials have been explored in conjunction with both gelatin and honey based wound dressings [27,28]. Both honey and gelatin have demonstrated antimicrobial and anti-inflammatory properties, which are essential to encouraging wounds to progress past the chronic inflammatory stage [27]. Gelatin based dressings, however, have demonstrated rapid, poorly controlled degradation rates [28], and neither honey nor gelatin-based matrices have been commercialized. The hybrid-scale fibers of the matrix discussed in this review have demonstrated the benefit of utilizing fibers of different sizes, allowing for controlled neo-vascularization and controlled absorption rates [17].

While initial clinical experience with the synthetic hybrid-scale fiber matrix has demonstrated positive clinical outcomes to date, several limitations are associated with the design of the clinical studies. Given the recent translation of hybrid-scale fiber matrix products into clinical use and practice, the level of clinical evidence available to date is limited to prospective and retrospective clinical studies, clinical case series, and case reports. No randomized controlled clinical trials have yet been completed examining the use of hybrid-scale fiber matrices as compared to existing human allogenic or xenogenic wound care products. Future clinical studies are therefore warranted in order to further elucidate the clinical and potential economic benefits associated with electrospun matrices above existing solutions. Successful completion of such comparative randomized clinical trials may solidify the role and value of electrospun matrices in both clinical and surgical settings.

## 6. Conclusions

This review represents a collective and comprehensive examination of a synthetic hybrid-scale fiber matrix in clinical practice and offers initial demonstration of the product’s mechanism of action and efficacy across multiple use cases in the wound care and surgical setting. Available clinical data confirms the versatility of the electrospun hybrid-scale fiber matrix across multiple wound types and clinical indications, and across multiple functions in the wound healing process. The evaluation provides evidence to suggest that synthetic electrospun matrices, in particular those that possess resorbable hybrid-scale fibers in the sub-micron to micron range, offer a unique alternative to existing biologic human allogenic and xenogenic wound care products.

## Figures and Tables

**Figure 1 bioengineering-10-00009-f001:**
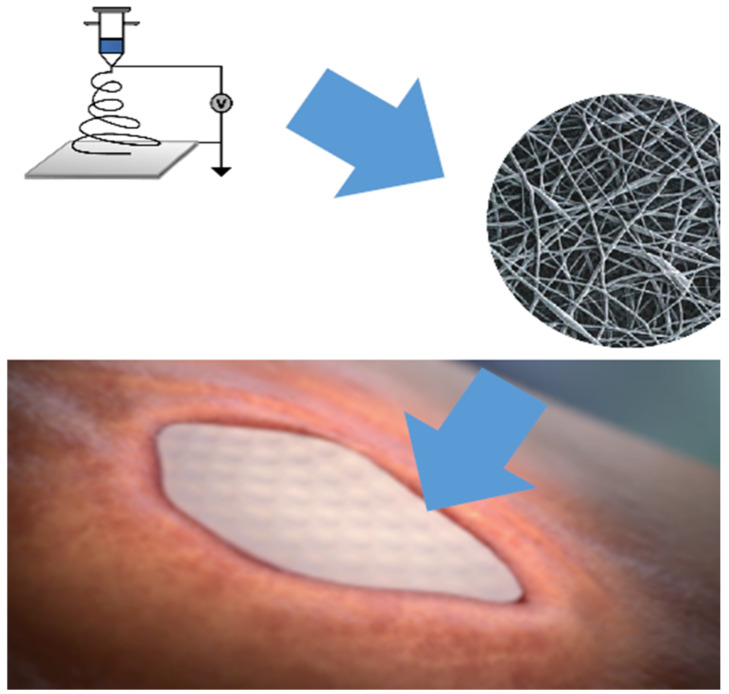
A graphical abstract demonstrating the electrospinning process and the microscopic appearance of the synthetic hybrid-scale fiber matrix, as well as its appearance in the wound bed.

**Figure 2 bioengineering-10-00009-f002:**
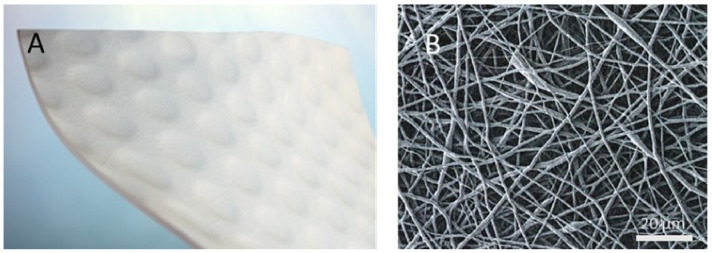
Synthetic hybrid-scale fiber matrix—(**A**) representative image of gross appearance and (**B**) scanning electron microscopy image.

**Figure 3 bioengineering-10-00009-f003:**
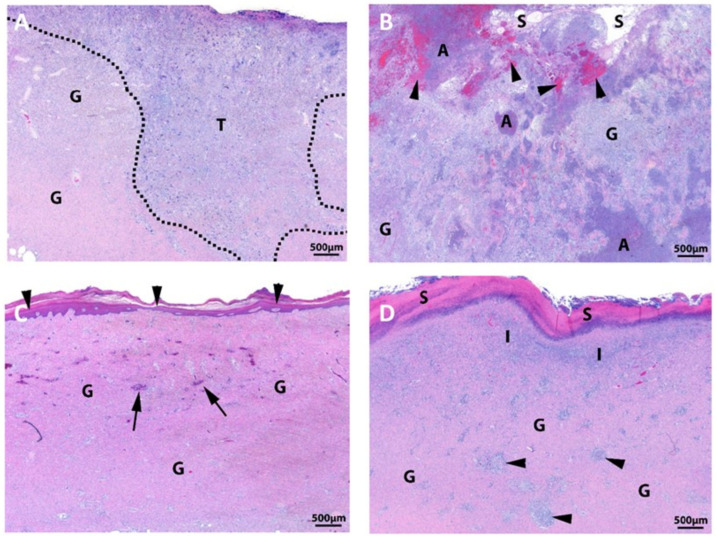
Histology images of full-thickness wounds stained with hematoxylin and eosin (H&E) at days 15 (**A**,**B**) and 30 (**C**,**D**) treated with the synthetic matrix (**A**,**C**) or the bilayer xenograft (**B**,**D**). Results showed that the synthetic matrix had less inflammation and faster filling of the wound bed with granulation tissue compared to the xenograft matrix. G—granulation tissue; T—area of granulation tissue containing inflammation and wound matrix; A—area of abscessation; S in (**B**)—seroma (edema) formation; S in (**D**)—serocellular debris; I—inflammation within granulation tissue; arrowheads in (**B**)—hemorrhage within wound bed; arrowheads in (**c**)—epithelium; arrowheads in (**D**)—small microgranulomas; arrows—blood vessels.

**Figure 4 bioengineering-10-00009-f004:**
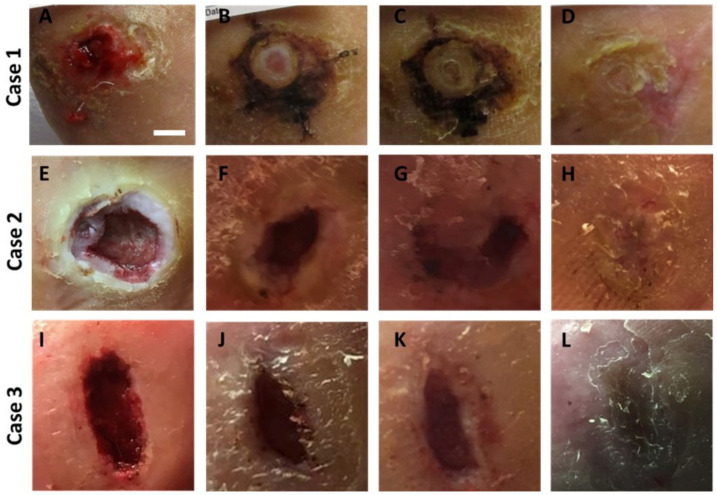
Representative images of progressive healing following treatment with the synthetic hybrid-scale fiber matrix. Case 1 diabetic foot ulcer at (**A**) week 0, (**B**) week 2, (**C**) week 4, and (**D**) complete healing at week 6. The scale bar represents 1 cm. Case 2 diabetic foot ulcer at (**E**) week 0, (**F**) week 5, (**G**) week 8, and (**H**) complete healing at week 12. Case 3 diabetic foot ulcer at (**I**) week 0, (**J**) week 1, (**K**) week 2, and (**L**) complete healing at week 4.

**Figure 5 bioengineering-10-00009-f005:**
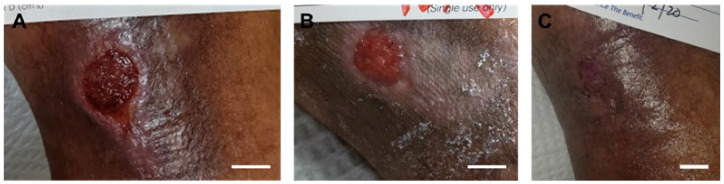
Treatment of a representative 5-year venous leg ulcer using the synthetic hybrid-scale fiber matrix. (**A**) Week 0, (**B**) week 5, and (**C**) complete healing at week 24. Scale bars represent 1 cm.

**Figure 6 bioengineering-10-00009-f006:**
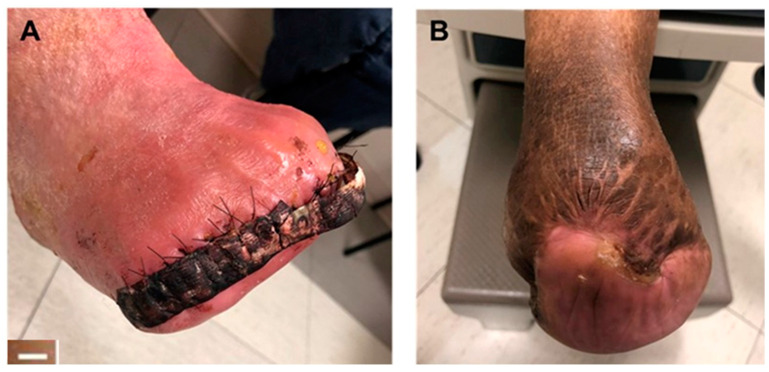
(**A**) A representative wound after TMA surgery. A strip of the synthetic hybrid-scale fiber matrix was applied over the closed wound. (**B**) A representative image of a healed TMA wound after treatment with the synthetic matrix. Scale bar represents 1 cm.

**Figure 7 bioengineering-10-00009-f007:**
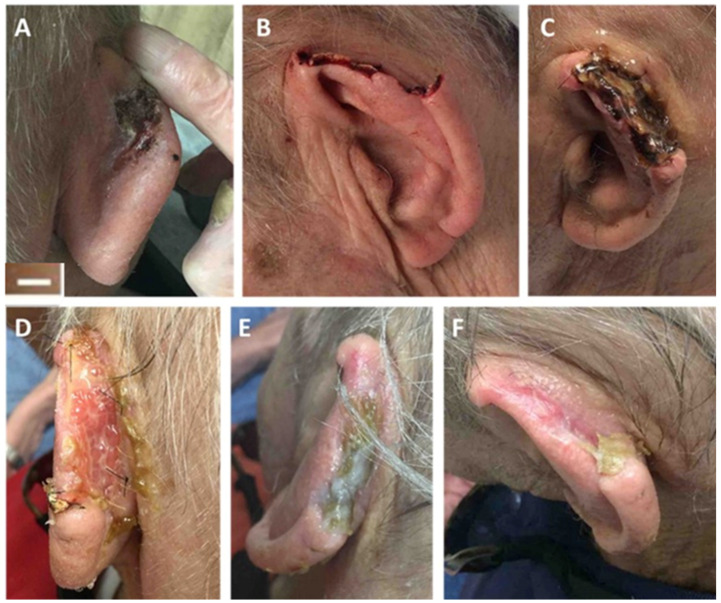
A representative Post-Mohs wound that healed over time following treatment with the synthetic hybrid-scale fiber matrix. (**A**) Squamous cell carcinoma (SCC) at the left posterior helix, (**B**) the resultant Mohs defect post-micrographic surgery, and (**C**) 2 weeks, (**D**) 3 weeks, (**E**) 5 weeks, and (**F**) 7 weeks after application of the synthetic hybrid-scale fiber matrix. Complete re-epithelialization was observed at 7 weeks. Scale bar represents 1 cm.

**Figure 8 bioengineering-10-00009-f008:**
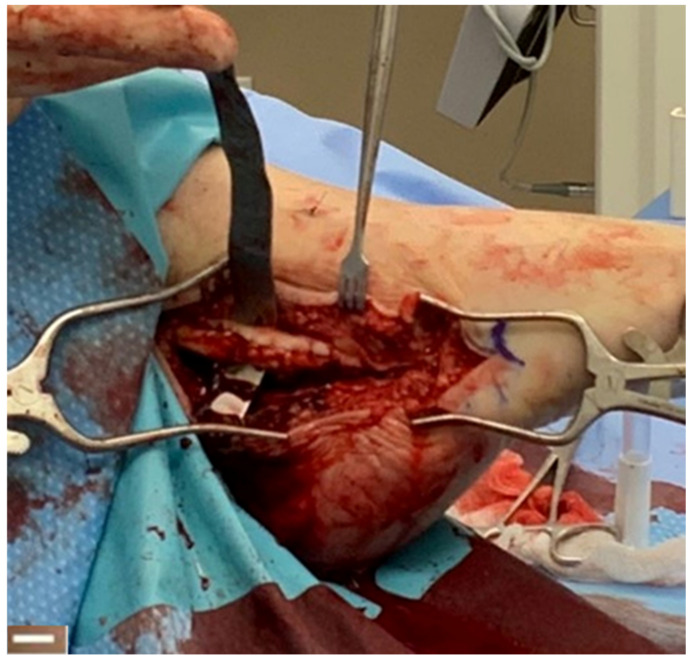
Tendon repair using the synthetic hybrid-scale fiber matrix. Scale bar represents 1 cm.

**Table 1 bioengineering-10-00009-t001:** Select available wound healing matrices.

Product	Supplier Information	Material Source (Allograft, Xenograft, or Synthetic)	Composition (Bovine Collagen, PGLA, etc.)	Clinical Applications (or Other Columns)?
Apligraf^®^	Organogenesis, Canton, MA, USA	Allograft	Neonatal foreskin-derived keratinocytes and fibroblasts with bovine Type I collagen	Management of serious wounds (i.e., ulcers)
DermACELL	Stryker, Kalamazoo, MI, USA	Allograft	Human tissue matrix	Management of serious wounds
Dermagraft^®^	Organogenesis, Canton, MA, USA	Allograft	Human fibroblasts seeded on polyglactin scaffold	Management of serious wounds
EpiFix	MiMedx, Marietta, GA, USA	Allograft	Dehydrated human amnion/chorion membrane	Management of serious wounds
Grafix^®^	Osiris Therapeutics, Inc, Columbia, MD, USA	Allograft	Cryopreserved human placental membrane	Management of serious wounds
Hyalomatrix	Anika Therapeutics, Bedford, MA, USA	Synthetic	Hyaluronic acid (HA) in fibrous form with an outer layer comprised of a semipermeable silicone membrane	Management of serious wounds
Integra Bilayer Wound Matrix Dressing	Integra, Plainsboro, NJ, USA	Xenograft	Cross-linked bovine tendon collagen and glycosaminoglycan and a semi-permeable polysiloxane (silicone layer)	Management of serious wounds
MicroLyte AG	Imbed Bioscience, Madison, WI, USA	Synthetic	Bioresorbable polyvinyl alcohol with a polymeric surface coating containing ionic and metallic silver	Management of minor (cuts, abrasions, etc.) and serious wounds
NovoSorb BTM	PolyNovo, Port Melbourne, Australia	Synthetic	Polyurethane	Management of serious wounds
Oasis^®^ Wound Matrix	Smith & Nephew, Fort Worth, TX, USA	Xenograft	Porcine-derived extracellular matrix	Management of serious wounds
TheraSkin^®^	Misonix, Farmingdale, NY, USA	Allograft	Human split thickness skin	Management of serious wounds

**Table 2 bioengineering-10-00009-t002:** Summary of clinical outcomes using the synthetic hybrid-scale fiber matrix.

Clinical Indication	# of Wounds	Patient Demographics	Treatment Method	Outcomes	Adverse Events	Reference
Chronic wounds (DFUs, VLUs, PUs, traumatic and postsurgical wounds, non-venous vascular wounds, necrotic wounds)	82	48% male; average patient age 72 years; average wound age 36 weeks; average wound surface area 3.4 cm^2^	Multiple applications of the synthetic hybrid-scale fiber matrix as needed for up to 12 weeks	85% complete wound closure at 12 weeks and significant reduction in local inflammation	None	[31]
Recalcitrant neuropathic foot ulcers	4	100% male; patient age range 67–73 years	Weekly, or as appropriate, treatment with synthetic hybrid-scale fiber matrix followed by adjunctive therapy	75% complete wound closure and successful limb preservation	None	[32]
DFUs, VLUs, TMAs, PUs, partial ray amputation, neuropathic ulcers	23	80% male, average patient age 63.7 years	Average number of applications was 1.2	96% wound closure at 95.1 days. Some wounds were also treated with NPWT and/or STSG	Wound dehiscence (1)	[33]
DFUs	24	90% male; average patient age 55 years; average ulcer duration 16 weeks; average ulcer surface area 4.4 cm^2^	Weekly, or as appropriate, treatment with synthetic hybrid-scale fiber matrix for up to 12 weeks	75% complete wound closure at 12 weeks	None due to synthetic matrix	[34]
Chronic wounds (DFU, VLU, PUs, Charcot foot deformity)	5	80% male; average patient age 66 years; average ulcer duration 51 months	Multiple applications of the synthetic hybrid-scale fiber matrix as needed in conjunction with NPWT	Formation of granulation tissue, coverage over exposed structures, and reduction in wound size	None	[35]
PUs	11	64% male; average patient age 55 years;	Single application of synthetic hybrid-scale fiber matrix as a foundation for rotational skin flap	Successful granulation tissue formation and preparation of wound site for flap reconstruction, with eventual wound closure rate of 90.9%	None	[36]
Chronic wounds (DFUs, VLUs)	23	60% male; average patient age 68 years; average ulcer duration 16 months	Weekly, or as appropriate, treatment with synthetic hybrid-scale fiber matrix	96% complete wound closure at 21 weeks	None due to synthetic matrix	[37]
Transmetatarsal amputations, Lisfranc amputation, Metatarsal/partial ray amputations	9	56% male. Patient age rand 52–68 years	Single application of synthetic hybrid-scale fiber matrix	78% wound closure. The synthetic hybrid-scale fiber matrix was utilized in conjunction with NPWT, STSG, and amniotic tissue.	Wound dehiscence (1), Infection (1)	[38]
TMA wounds	20	85% male; average patient age 62 years	10 wounds treated with synthetic hybrid-scale fiber matrix to augment closure of the suture line and 10 control nonaugmented wounds with standard primary closure	80% complete wound closure following treatment with synthetic matrix; reduced time to healing (18%), compared to control	Wound dehiscence (5), limb loss (2)	[39]
Post-Mohs wounds	4	75% male; average patient age 78 years; average ulcer surface area 11.5 cm^2^	Multiple applications of the synthetic hybrid-scale fiber matrix as needed	100% complete wound closure in 8 weeks with no scars or skin deformities	None	[40]
Peroneal tendon healing	12	25% male; patient age range 18–75 years	Peroneal tendon repair augmented with synthetic hybrid-scale fiber matrix	Significant reduction in pain and rapid return to normal activity	None	[41]
Complex cutaneous wounds (calciphylaxis lesion, abdominal fistula lesion, necrotizing fasciitis lesion)	3	67% male; patient age range 30–54 years	Multiple applications of the synthetic hybrid-scale fiber matrix as needed in conjunction with NPWT	Significant re-epithelialization and healing of the wounds and economic cost savings	None	[42]
Traumatic crush injury wound	1	24-year-old male	Single application of synthetic hybrid-scale fiber matrix as a foundation for STSG	Successful granulation tissue formation and preparation of wound site for STSG	None	[43]
Hematomas	2	50% male, patient age ranges 59–82	Multiple applications of the synthetic hybrid-scale fiber matrix as needed. Used in conjunction with NPWT for one patient	100% wound closure at an average of 77 days post initial treatment	None	[44]

DFU—diabetic foot ulcer; HBOT—hyperbaric oxygen therapy; NPWT—negative pressure wound therapy; PUs—pressure ulcers; STSG—split thickness skin graft; TMA—transmetatarsal amputation; VLUs—venous leg ulcers.

## Data Availability

Not applicable.

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
