# Peer review of "Clinical Application of Bioresorbable, Synthetic, Electrospun Matrix in Wound Healing"

_bioengineering, 2022, doi:10.3390/bioengineering10010009_

Round 1

Reviewer 1 Report

The review by Matthew MacEwan et al. gives an overview on the advances in electrospun hybrid fiber matrix for clinical wound healing application. This manuscript was well organized and written. Some minor revisions should be performed before publication:

1. The title is a little bit confused. Please explain what the hybrid-scale means.

2. Abstract must be presented in a better and clear way. For instance, the Background part is too long, and it should be shorten.

3. The main reviews in the area of electrospinning-based strategies for wound healing application should be mentioned in the introduction as well. A justification for an additional review in this field should be given.

4. A graphical abstract that can reflect the scope of the review well is recommended to be drawn and presented in this paper, which will be helpful for understanding of readers.

5. What are the merits and demerits of employing PLGA (10:90) and PDO as wound dressing materials compared with the others like PLLA, PCL, collagen and alginate?

6. Some recent works in this area like 10.3390/nano12152560, 10.1016/j.apmt.2022.101542, 10.3390/nano10010175, and 10.3390/nano12050784 are missing, which should be discussed in the discussion section.

7. Some statements feel they are lacking references. Admittedly, some of these statement might be considered well known facts, the concepts mentioned might have been referenced previously or will be in the future, but it might still be pertinent add references next to these statements for new readers to the field, skim readers or people who don't necessarily want to go looking for the relevant reference.

8. Some grammar and typo mistakes should be double checked and revised in the whole manuscript. For instance, lines 147, 206, and 377, etc.

Reviewer 2 Report

The manuscript is well written and comprehensive. I have a few things to consider. The  phrase "Error! Reference source not found" appeared six times throughout the text, please added the correct word for the sentence.    To keep only the title of Table 2, the abbreviations will appear in the note of this table. Please check the references "in press", there were changes in the title after  the publication. 

The photographs belong the articles mentioned in the review, if yes, please following the role. Some photos showed scale bars, please keep it in all photos. On page 9, lines 255 to 258, it's the same information that's in the legend.

There are some references to belong to a conference or symposium, I thinks is not adequate to scientific article. 
